# CRANE: Reasoning with constrained LLM generation

**Debangshu Banerjee** [1] [*]   **Tarun Suresh** [1] [*]   **Shubham Ugare** [1]   **Sasa Misailovic** [1]   **Gagandeep Singh** [1]

## Abstract

Code generation, symbolic math reasoning, and other tasks require LLMs to produce outputs that are both syntactically and semantically correct. Constrained LLM generation is a promising direction to enforce adherence to formal grammar, but prior works have empirically observed that strict enforcement of formal constraints often diminishes the reasoning capabilities of LLMs. In this work, we first provide a theoretical explanation for why constraining LLM outputs to very restrictive grammars that only allow syntactically valid final answers reduces the reasoning capabilities of the model. Second, we demonstrate that by augmenting the output grammar with carefully designed additional rules, it is always possible to preserve the reasoning capabilities of the LLM while ensuring syntactic and semantic correctness in its outputs. Building on these theoretical insights, we propose a reasoning-augmented constrained decoding algorithm, CRANE, which effectively balances the correctness of constrained generation with the flexibility of unconstrained generation. Experiments on multiple open-source LLMs and benchmarks show that CRANE significantly outperforms both state-of-the-art constrained decoding strategies and standard unconstrained decoding, showing up to 10% points accuracy improvement over baselines on challenging symbolic reasoning benchmarks GSM-symbolic and FOLIO.

## 1. Introduction

Transformer-based large language models (LLMs) are widely used in AI systems that interact with traditional software tools like Python interpreters (OpenAI, 2024; Chen et al., 2023) for code generation (Suresh et al., 2024a;b;

Jiang et al., 2024), logical solvers (Pan et al., 2023; Olausson et al., 2023), and theorem provers (Wu et al., 2022; Yang et al., 2023). These tools impose specific syntactic and semantic constraints on their inputs, requiring LLMs to produce outputs in the correct format. For instance, if an LLM provides output to a specific logical solver (Han et al., 2024), the output must be parsable by that solver. Similarly, Wolfram Alpha (wolfram, 2024) translates user queries about mathematical problems into a domain-specific language (DSL) to utilize symbolic solvers. However, as highlighted in recent studies (Ugare et al., 2024b; Lundberg et al., 2023; Poesia et al., 2022), pre-trained LLM outputs do not always comply with downstream tools' input requirements. Constrained decoding algorithms (Ugare et al., 2024b; Poesia et al., 2022) address this issue by projecting the LLM output onto user-specified formal constraints (e.g., syntactic rules defined by a context-free grammar $G$), thereby ensuring that the input requirements of downstream tasks are satisfied.

As illustrated in Fig. 1, constrained decoding improves the syntactic correctness of LLM outputs (e.g., generating a well-formed mathematical expression). However, it does not guarantee functional correctness (e.g., ensuring the expression correctly answers the user's query). Recent works such as Tam et al. (2024) have empirically observed that imposing constraints on LLM outputs can, in some cases, reduce functional correctness for specific tasks. Tam et al. (2024) attributes this reduction in functional accuracy to a decline in the LLM's reasoning capabilities under constrained decoding. This observation raises the following open questions:

- **RQ1**: Do LLMs truly lose reasoning capabilities under constrained decoding?
- **RQ2**: How can we leverage the benefits of constrained decoding in reducing syntax errors while preserving the unconstrained reasoning capabilities of LLMs?

**Key Challenges:** First, we need to formally identify the root cause of the reduction in functional accuracy of end-to-end systems when a pre-trained LLM operates under constrained generation. Unlike the empirical observations in (Tam et al., 2024), we seek a formal justification for this reduction that is not limited to specific LLMs used in experiments but extends to any LLM, including more

---
[*]Equal contribution [1]Department of Computer Science, University of Illinois Urbana-Champaign, USA. Correspondence to: Debangshu Banerjee <db21@illinois.edu>.

*Proceedings of the $42^{nd}$ International Conference on Machine Learning*, Vancouver, Canada. PMLR 267, 2025. Copyright 2025 by the author(s).

*Figure 1.* An example from the GSM-symbolic dataset (variables in blue) where unconstrained generation produces syntactically incorrect output, while constrained generation provides a syntactically valid but incorrect answer. CRANE, however, generates a correct answer.

powerful ones developed in the future.

Second, we must design cost-efficient decoding strategies that address the shortcomings of existing constrained decoding methods while improving functional accuracy. In this work, we do not consider task-specific fine-tuning of LLMs, as fine-tuning for each task is compute-intensive. Unlike constrained decoding, fine-tuning does not guarantee that the LLM output adheres to formal constraints.

**Contributions:** We make the following contributions to improve the functional accuracy of the end-to-end system:

- We theoretically show that LLMs with a constant number of layers, which are known to be capable of simulating $n$ steps of any given Turing machine $M$ with $O(n)$ reasoning steps (Merrill & Sabharwal, 2024), can only solve problems within a relatively restrictive circuit complexity class when constrained to generate outputs that always conform to a restrictive grammar $G$ defining only the valid output strings. This demonstrates that, for restrictive grammar, constrained decoding reduces the problem-solving capabilities of LLMs.
- We theoretically show that the loss of expressivity of LLMs under constrained decoding arises because the output grammar $G$ is too restrictive to accommodate the intermediate reasoning steps required to compute the answer. We further demonstrate that augmenting the grammar $G$ with specific additional production rules enables the LLM to generate the intermediate reasoning steps while ensuring that the final output always adheres to the intended output structure. With the augmented grammar $G_a$, the LLM retains its expressivity under constrained decoding.
- We propose a simple and cost-efficient decoding strategy, CRANE (**C**onstrained **R**easoning **A**ugmented **Gen**eration). CRANE effectively alternates between unconstrained generation for reasoning and constrained gen-

eration for producing structurally correct outputs. This allows the model to produce syntactically valid outputs while enabling the LLM to reason. Our detailed experiments on multiple open-source LLMs and benchmarks demonstrate that CRANE significantly outperforms both SOTA constrained decoding strategies and standard unconstrained decoding, showing up to a 10% improvement over baselines on challenging symbolic reasoning benchmarks GSM-symbolic (Mirzadeh et al., 2024)) and FOLIO (Han et al., 2024).

Next, we provide the notations and necessary background on constrained decoding, including the definition of Turing machines and relevant circuit complexity classes.

## 2. Preliminaries

**Notations:** In the rest of the paper, we use small case letters ($x$) for constants, bold small case letters ($\boldsymbol{x}$) for strings, capital letters $X$ for functions, $\cdot$ for string concatenation, $|\boldsymbol{x}|$ to denote the length of the string $\boldsymbol{x}$. We use LLM to refer to transformer-based LLMs with a fixed number of layers.

### 2.1. Constrained LLM Decoding

Autoregressive language models $\mathcal{L}$ decode output iteratively by generating tokens from a probability distribution over the vocabulary $V$. The distribution is derived by applying the softmax function to the model's scores $\mathcal{S}$. Common decoding methods include greedy decoding, temperature sampling, and beam search. Constrained LLM decoding extends this process by excluding specific tokens at certain positions, such as avoiding harmful words or adhering to a user-defined output grammar for languages like JSON or SQL (Poesia et al., 2022; Ugare et al., 2024c; Willard & Louf, 2023). At each decoding step, a binary mask $m \in \{0,1\}^{|V|}$, generated by a function $f_m$, specifies valid tokens

($m_i = 1$) and excluded tokens ($m_i = 0$). Decoding is then performed on the masked probability distribution $m \odot softmax(\mathcal{S})$, where $\odot$ denotes element-wise multiplication.

## 2.2. Deterministic LLM Decoding

CRANE is compatible with various decoding strategies, both constrained and unconstrained, allowing the output of $\mathcal{L}$ to be stochastic. However, following existing works (Hahn, 2020; Merrill & Sabharwal, 2023; Li et al., 2024) and for simplicity in the theoretical setup in Section 3, we assume that the output of $\mathcal{L}$ on any input string $\boldsymbol{x}$ is deterministic in both constrained and unconstrained settings.

Similar to prior works (Merrill & Sabharwal, 2023; 2024), we model a single autoregressive step as a deterministic function $\mathcal{L}_f$ that predicts the next token given a specific input. Formally,

**Definition 2.1** (Deterministic LLM Step). A single autoregressive step of an LLM is modeled as a deterministic function $\mathcal{L}_f : V^* \to V$, where $V$ is the finite vocabulary and $V^*$ represents the set of all finite strings over $V$. For an input string $\boldsymbol{x} \in V^*$, the LLM predicts the next token $\mathcal{L}_f(\boldsymbol{x})$.

**Definition 2.2** (Deterministic Unconstrained Decoding). For an input string $\boldsymbol{x}$, the deterministic output string $\boldsymbol{y}$ selected from the output distribution of a LLM using a decoding algorithm (e.g., greedy decoding) is denoted as $\boldsymbol{y} = \mathcal{L}(\boldsymbol{x})$ where $\mathcal{L} : V^* \to V^*$. $\mathcal{L}(\boldsymbol{x})$ is the most likely output sequence according to learned distribution on $\boldsymbol{x}$.

The output $\boldsymbol{y} = \mathcal{L}(\boldsymbol{x})$ is computed iteratively with $|\boldsymbol{y}|$ autoregressive steps defined by $\mathcal{L}_f$. For each $1 \leq i \leq |\boldsymbol{y}|$, and the recurrence relation $\mathcal{L}_f^{(i)}(\boldsymbol{x}) = \mathcal{L}_f^{(i-1)}(\boldsymbol{x}) \cdot \mathcal{L}_f(\mathcal{L}_f^{(i-1)}(\boldsymbol{x}))$ where $\mathcal{L}_f^{(0)}(\boldsymbol{x}) = \boldsymbol{x}$ and $\cdot$ denotes string concatenation. Here, $\boldsymbol{x} \cdot \boldsymbol{y} = \mathcal{L}_f^{|\boldsymbol{y}|}(\boldsymbol{x})$. Similarly, under constrained decoding with a grammar $G$ we define:

**Definition 2.3** (Deterministic Constrained Decoding under Grammar). Under constrained decoding with a formal grammar $G$, the output string $\boldsymbol{y}_G$ is selected from the constrained output distribution and is denoted as $\boldsymbol{y}_G = \mathcal{L}_G(\boldsymbol{x})$. The output of $i$-th constrained autoregressive step with $G$ is $\boldsymbol{x} \cdot \boldsymbol{y}_G^{(i)} = \mathcal{L}_G^{(i)}(\boldsymbol{x})$ and $\boldsymbol{x} \cdot \boldsymbol{y}_G = \mathcal{L}_G^{(|\boldsymbol{y}|)}(\boldsymbol{x})$.

The constrained output $\boldsymbol{y}_G$ is always in the grammar $\boldsymbol{y}_G \in L(G)$ where $L(G)$ is the language defined by $G$. For sound-constrained decoding algorithms, if the unconstrained output $\boldsymbol{y} = \mathcal{L}(\boldsymbol{x})$ in the grammar $\boldsymbol{y} \in L(G)$, the constrained output remains unchanged, i.e., $\mathcal{L}(\boldsymbol{x}) = \mathcal{L}_G(\boldsymbol{x})$.

## 2.3. LLM Expressivity

We discuss the notations and background related to Turing machines, and relevant uniform circuit complexity classes. Turing machines are popular mathematical computation

models used to analyze resource requirements (e.g. time and space complexity) and the hardness of computation problems. Formally, a Turing machine is defined as:

**Definition 2.4** (Turing Machine). A Turing machine $M$ with $k$ work tapes and an output tape is a 8-tuple

$$M = \langle \Sigma, \Gamma, k, b, Q, q_0, \delta, F \rangle,$$

where $\Sigma$ is the finite input alphabet, $\Gamma$ is the finite tape alphabet with $\Sigma \subseteq \Gamma$, $b \in \Gamma \setminus \Sigma$ is a special blank symbol, $Q$ is a finite set of states, $q_0 \in Q$ is the initial state, $\delta : (Q \setminus F) \times \Gamma^{k+2} \to Q \times \Gamma^{k+1} \times \{0, +1, -1\}^{k+2}$ is the transition function (where $-1, 1, 0$ represent moving the tape head left, right, or staying in place, respectively), and $F \subseteq Q$ is the set of halting states.

Let $\Sigma^*$ denote the set of all finite strings over the input alphabet $\Sigma$. Given an input string $\boldsymbol{s} \in \Sigma^*$, the computation of $M$ on $s$ is a sequence of configurations starting from the initial configuration. Each configuration $\gamma$ is a tuple containing the current state $q \in Q$, the contents of the input tape, the $k$ work tapes, the output tape, and the current head positions of all $k + 2$ tapes. For each configuration, $\gamma_i$ ($i \in \mathbb{N}$), the transition function $\delta$ computes the next configuration $\gamma_{i+1}$ based on the current state $q$ and the values on the $k + 2$ tapes at the current head positions. It updates the head positions, writes to the output tape (possibly leaving it unchanged if no new symbol is written), and advances to the next configuration. For each $i$, computation of $\gamma_{i+1}$ from $\gamma_i$ defines a single step of the Turing machine.

The computation of $M$ on input $\boldsymbol{s}$ halts if $M$ reaches a halting state $q \in F$. If $M$ halts, the output corresponding to $\boldsymbol{s}$ is written on the output tape. Additional details about the computation of the Turing machine are in Appendix A.

Before discussing existing expressivity results for constant-layer LLMs, we briefly introduce relevant uniform constant-depth circuit complexity classes, e.g. logspace uniform-$TC^0$, which provide an upper bound on the computational power of LLMs that do not employ reasoning steps, as seen in methods like Chain-of-Thought (Wei et al., 2022).

**Definition 2.5** (Boolean Circuit). A Boolean circuit is a computational model for evaluating Boolean functions over fixed-length binary strings. It is represented as a directed acyclic graph (DAG), where the leaf nodes correspond to input binary variables or their negations, and the internal nodes perform operations from a predefined set of operations $\mathcal{B}$ (e.g., AND ($\wedge$), OR ($\vee$), etc.). One or more marked nodes in the graph represent the circuit's output.

The structure of the DAG specifies the computation of the Boolean function by propagating input values through the graph. The complexity of a circuit is determined by its size (the number of nodes) and depth (the longest path in the graph). Since a single circuit only defines a boolean

function for fixed-length inputs, a family of circuits is required—one for each input length—to characterize a computational problem where input lengths vary. Unlike Turing machines, whose computation does not depend on input length, circuit families have a separate circuit for each input length, which can differ entirely. This non-uniformity can lead to degenerate cases where non-uniform circuit families solve undecidable problems (Arora & Barak, 2009). To address this issue, complexity theorists enforce uniformity conditions, requiring circuits for different input sizes to be related, resulting in uniform circuit families. For further details and formal definitions of circuit classes, refer to (Arora & Barak, 2009). In this work, we focus on constant-depth, polynomial-sized logspace-uniform threshold circuits ($TC^0$), where $\mathcal{B}$ contains only threshold gates (a formal definition is in Appendix B).

## 3. Expressivity of Constrained Decoding

First, we show that any constant-layer LLM $\mathcal{L}$ under constrained decoding loses expressivity. We identify the class of problems and the corresponding output grammars $G$ such that when imposed on the outputs of any constant-layer LLM, the problems cannot be solved unless there is a collapse in fundamental complexity classes that are widely believed to be unequal (e.g., $TC^0 \neq NL$)[1].

### 3.1. Limitation of Constrained Decoding

Next, we present the high-level idea behind Proposition 3.1 that shows the limitation of constrained LLM decoding when the output grammar is too restrictive. We consider problems where the number of possible outputs is finite, and thus the set of all possible outputs $O$ can be expressed as a simple regular language. Consequently, $G_c$ that encodes the output set $O$, i.e., $O = L(G_c)$, where $L(G_c)$ denotes the language defined by the grammar $G_c$. For instance, any decision problem (yes/no answer) such as st-connectivity that asks for vertices s and t in a directed graph, if t is reachable from s can be answered within a single-bit output i.e. $L(G_c) = \{0, 1\}$. This implies that constrained decoding with the output grammar $G_c$ allows only a single autoregressive step for any $\mathcal{L}$ on all inputs.

A series of existing works (Hahn, 2020; Hao et al., 2022; Merrill et al., 2022; Merrill & Sabharwal, 2023) establish that, under suitable assumptions, a single autoregressive step on an input with length $n$ for any constant-depth LLM can be represented as a constant-depth circuit. Since, for decision problems, the constrained decoding step permits only a single autoregressive step, any LLM can only solve problems within the corresponding circuit complexity class. We build on the most recent result from (Merrill & Sabharwal, 2023), which shows that a single autoregressive step of any

LLM with a constant number of layers on an input of length $n$ can be simulated by a logspace-uniform constant-depth threshold circuit family. This result allows the LLM to use floating-point numbers with $\log(n)$ precision when processing inputs of size $n$, ensuring that the precision scales with $n$ and preventing floating-point representation issues for large $n$. We denote such LLMs as log-precision LLMs.

Let $\boldsymbol{x} \cdot \boldsymbol{y}^{(i)}$ denote the output after the $i$-th autoregressive step of an LLM $\mathcal{L}$ under constrained decoding with an output grammar $G$ on input $\boldsymbol{x}$. Then, we have $\boldsymbol{x} \cdot \boldsymbol{y}^{(i)} = \mathcal{L}_G^{(i)}(\boldsymbol{x})$, and for any $i$, $\boldsymbol{y}^{(i)}$ is always a valid prefix of a string in $L(G)$, i.e., there exists a (possibly empty) string $\boldsymbol{\alpha}^{(i)}$ such that $\boldsymbol{y}^{(i)} \cdot \boldsymbol{\alpha}^{(i)} \in L(G)$. Now, for any output grammar $G_c$ where the output set $O = L(G_c)$ is finite, we show that the output $\mathcal{L}_{G_c}(\boldsymbol{x})$ for any input $\boldsymbol{x}$ of size $|\boldsymbol{x}| = n$ can be computed using constant-depth threshold circuits.

**Proposition 3.1.** *For any log-precision LLM $\mathcal{L}$ with constant layers there exists a logspace-uniform thershold circuit $Th_n$ such that $\mathcal{L}_{G_c}(\boldsymbol{x}) = Th_n(\boldsymbol{x})$ holds for all inputs $\boldsymbol{x}$ with size $|\boldsymbol{x}| = n$ and $n \in \mathbb{N}$.*

**Proof:** The formal proof is in Appendix C.

From Proposition 3.1, it follows that for any decision problem under constrained decoding, an LLM can only solve problems within the logspace-uniform $TC^0$ class (constant-depth threshold circuits). Consequently, any decision problem believed to lie outside this class cannot be solved under constrained decoding. The previously mentioned st-connectivity problem is known to be $NL$-complete (Arora & Barak, 2009). This implies that unless $TC^0 = NL$, no LLM under constrained decoding can solve st-connectivity. Additionally, (Li et al., 2024; Merrill & Sabharwal, 2024) show that given any Turing machine $M$ there exists a log-precision LLM with a constant number of layers that can simulate $O(t(n))$ steps of $M$ using $O(t(n))$ autoregressive steps, where $t(n)$ denotes a polynomial in the input size $n$.

**Lemma 3.2.** *For any Turing machine $M$ with tape alphabet $\Gamma$, there exists a constant depth LLM $\mathcal{L}_M$ with finite vocabulary $\Gamma \subseteq V_M$ and log-precision that can simulate $t(n)$ steps of $M$ with $t(n)$ autoregressive steps.*

**Proof:** The proof follows from Theorem 2 in (Merrill & Sabharwal, 2024) further details in Appendix C.

Proposition 3.1 and Lemma 3.2 together imply that there exist problems, such as st-connectivity, an LLM can solve that in an unconstrained setting but cannot be solved under constrained decoding (unless logspace-uniform $TC^0 = NL$).

### 3.2. Reasoning with Augmented Grammar

The reduction in LLM expressivity under constrained decoding, as established in Proposition 3.1, arises primarily because the language of all valid output strings, $L(G_c)$,

---

is too restrictive and does not permit large (non-constant) reasoning chains. This naturally leads to the question of whether it is possible to augment any output grammar $G$ with additional production rules to construct an augmented grammar $G_a$ that can accommodate reasoning steps while preserving the expressivity of $\mathcal{L}$ even under constrained decoding. At the same time, $G_a$ should remain nontrivial—meaning it should not accept all possible strings, as in the unconstrained setting—so that it aligns with the practical objective of constrained decoding: guiding the LLM to generate syntactically and semantically valid outputs.

To achieve this, we enforce that the augmented grammar $G_a$ always follows the structure $G_a \rightarrow RG$, where the nonterminal symbol $R$ captures the reasoning steps, and $G$ represents the final output. This guarantees that for any string $s \in L(G_a)$, the final answer $a$ extracted from $s = r \cdot a$ always belongs to the original output grammar $G$, i.e., $a \in L(G)$, with $r$ serving as the reasoning sequence leading up to the final output.

Formally, we show that for any Turing machine $M$ and a grammar $G$ containing all valid outputs of $M$, there exists an LLM $\mathcal{L}_M$ with a constant number of layers and log-precision, along with an augmented grammar $G_a$ in the specified format, such that $\mathcal{L}_M$ can simulate $t(n)$ steps of $M$ using $t(n)$ autoregressive steps under constrained decoding with $G_a$. Here $n \in \mathbb{N}$ and $t(n)$ is a polynomial over $n$. The augmented grammar $G_a$ may not be unique, and we provide one such construction.

At a high level, $\mathcal{L}_M$ simulates the Turing machine $M$ by computing the encoded representations $\overline{\gamma_i}$ of the machine's configurations $\gamma_i$ at each step $i$ and storing them within the reasoning component (i.e., the string $r$) of the output. During each autoregressive step, $\mathcal{L}_M$ generates the next configuration based on the transition function of $M$ and appends its encoding to the reasoning sequence. This process continues until $M$ reaches a halting state, at which point $\mathcal{L}_M$ produces the final output $a$, which belongs to $L(G)$. For any given $M$, we define the rules $R_M$ that can parse the encodings $\overline{\gamma}$ of all possible configurations $\gamma$. This ensures that the output $\mathcal{L}_{G_a}(x)$ represents the full reasoning-augmented sequence, i.e., $\overline{\gamma_1} \cdots \overline{\gamma_{t(n)}} \cdot M(x)$, where $M(x)$ is the final output of $M$ on input $x$ of size $n$ after $t(n)$ computational steps. The encodings $\overline{\gamma_1}, \ldots, \overline{\gamma_{t(n)}}$ correspond to the configurations $\gamma_1, \ldots, \gamma_{t(n)}$, as described below.

We begin by defining the vocabulary $V_M$ for $\mathcal{L}_M$, which contains all tape symbols $\Gamma$ of $M$ along with a finite set of auxiliary symbols $\overline{\gamma}$ that encode the corresponding configurations $\gamma$. Similar to prior works (Merrill & Sabharwal, 2024), each configuration encoding $\overline{\gamma}$ represents the current state $q$, the symbols at the current head position of $k + 2$ tapes (input, output and $k$ work tapes), and the head movement directions $\{0, +1, -1\}$ for each tape. Directions

$\{0, +1, -1\}$ denote either staying in place (0), moving left ($-1$), or moving right ($+1$) by a single position. Since the set of states $Q$, the tape alphabet $\Gamma$, and the number of tapes $k$ are all constants, the total number of possible encodings $\overline{\gamma}$ is also constant. Let $\overline{\Gamma}$ denote the set of all possible configuration encodings, i.e., $\overline{\Gamma} = \{\overline{\gamma_{(1)}}, \ldots, \overline{\gamma_{(l)}}\}$, where $l = |\overline{\Gamma}|$. Given $\overline{\Gamma}$ is finite and enumerable, we can define the rules of the augmented grammar $G_a$ accordingly as follows.

$$G_a \rightarrow R_M G; \quad R_M \rightarrow S R_M; \quad S \rightarrow \overline{\gamma_{(1)}} \mid \cdots \mid \overline{\gamma_{(l)}}$$

The set of reasoning strings in $L(R_M)$ essentially define a regular language over the configuration encodings $\overline{\Gamma}$. Let, for any input $x$ with size $n = |x|$ a given Turing machine $M$ halts and compute the output $M(x)$ in $t(n)$ steps that are polynomial in $n$. Then there exist $\mathcal{L}_M$ compute $M(x)$ with $t(n)$ autoregressive steps under constrained decoding with the augmented grammar $G_a \rightarrow R_M G$. Suppose, $\mathcal{L}_{M,G_a}(x)$ denotes the output of the LLM $\mathcal{L}_M$ on input $x$ under constrained decoding with grammar $G_a$ then

**Proposition 3.3.** *For any Turing machine $M$ with tape alphabet $\Gamma$, there exists a constant depth LLM $\mathcal{L}_M$ with finite vocabulary $\Gamma \subseteq V_M$ and log precision such that for any input $x$ with $|x| = n$, $\mathcal{L}_{M,G_a}(x) = r \cdot M(x)$ with $r \in V_M^*$ assuming $M$ halts on $x$ in $t(n)$ steps.*

**Proof:** The proof is in Appendix C.

## 4. CRANE Algorithm

Given any Turing machine $M$, Proposition 3.3 establishes that constrained decoding with the augmented grammar $G_a$ on a specific LLM $\mathcal{L}_M$ can simulate the computation of $M$. However, this result does not directly translate into a practical constrained decoding algorithm that preserves the expressivity of general LLMs. The construction assumes a specific LLM $\mathcal{L}_M$ with the vocabulary $V_M$ and knowledge of the particular Turing machine $M$ for defining the rules $R_M$. In practice, we require an efficient approach that can be applied to diverse open-source LLMs, various grammars, and different constrained decoding algorithms. Importantly, we know that enforcing the output grammar $G$ from the beginning can limit expressivity. Instead, we impose grammar constraints judiciously to avoid restricting the LLM's reasoning capabilities. For example, in the case of a reasoning-augmented output of the form $\overline{\gamma_1} \cdots \overline{\gamma_{t(n)}} \cdot M(x)$, we apply constrained decoding only from the $t(n) + 1$-th autoregressive step onward, ensuring that the reasoning process remains unrestricted while the final answer adheres to the desired grammar.

The primary challenge here is deciding when to transition between an unconstrained generation for reasoning and a

constrained generation. For instance, grammar for general-purpose programming languages such as Python can allow any text string at the start (e.g. program starting variable names) making it hard to detect the end of reasoning string. To avoid this, we augment the output grammar with specific delimiter symbols $S_1$ and $S_2$ that mark the start and end of the constrained generation. We incentivize the LLM to generate these delimiters via explicit instructions in the prompt and few-shot examples. This aligns with common general-purpose LLMs that already use specific delimiters such as backticks (```) for programs like python, SQL, and ($<<, >>$) to enclose math-expression blocks. This approach allows a simple and cost-efficient approach for detecting the transitions to and from constrained decoding. For the construction in the previous section, in this setup, we will generate the string $r \cdot S_1 \cdot M(x) \cdot S_2$ where the reasoning $r$ is generated unconstrained and the LLM moves to constrained mode after seeing the symbol $S_1$. However, in practical cases, the delimiters may be generated multiple times (ie. for intermediate operations), even during the reasoning step. Therefore, upon encountering the end symbol $S_2$, we switch back to unconstrained generation to avoid unnecessarily restricting the output.

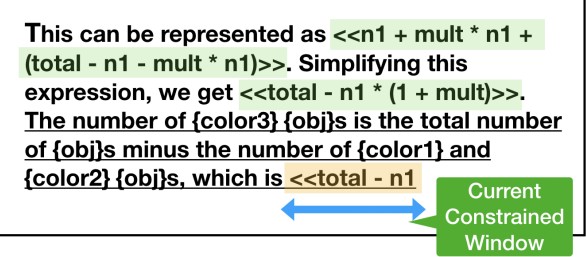

*Figure 2.* CRANE adaptively switches between constrained LLM generation and unconstrained LLM generation based on start and end delimiters (in this example $<<$ and $>>$). Using these delimiters, CRANE dynamically tracks which windows (highlighted in the figure) of the LLM generation constraints should be applied to.

We implement our approach into the CRANE algorithm (Algo 1), which extends standard autoregressive LLM generation. CRANE takes an arbitrary LLM, constrained decoding algorithm (denoted as CSD), output grammar $G$, and symbols $S_1$ and $S_2$ as input. It first initializes CSD with $G'$, the output grammar augmented with $S_1$ and $S_2$. CRANE starts in unconstrained generation and maintains a pointer that marks the start of the current window of LLM generation following the last constrained generation. In each iteration, the algorithm checks if $S_1$ is present in the current generation window `currGen`, which is the portion of the sequence from the current pointer position onwards. If $S_1$ is detected, CRANE switches to constrained generation mode. In this mode, the current constrained window (the portion of `currGen` that is in $G'$) is extracted, and the next token is sampled based on the constraints defined by the CSD.

---

**Algorithm 1** CRANE Algorithm

1: **Input:** LLM, tokens, CSD (constrained decoder), G (output grammar), $S_1$ (start delimiter), $S_2$ (end delimiter)
2: **Output:** Output string
3: $G' \leftarrow S_1 G S_2$
4: CSD.INITIALIZE($G'$)
5: `pointer` $\leftarrow$ len(tokens)
6: `isConstrained` $\leftarrow$ False
7: **while** True **do**
8:     `currGen` $\leftarrow$ detokenize(tokens[`pointer` :])
9:     **if** $S_1 \in$ `currGen` **then**
10:        `isConstrained` $\leftarrow$ True
11:     **else**
12:        `isConstrained` $\leftarrow$ False
13:     **if** `isConstrained` **then**
14:        constrained $\leftarrow$ extractConstrained(`currGen`)
15:        $t_i \sim$ LLM(tokens) $\odot$ CSD(constrained)
16:     **else**
17:        $t_i \sim$ LLM(tokens)
18:     tokens $\leftarrow$ tokens $+ t_i$
19:     **if** $t_i =$ EOS **then**
20:        **break**
21:     **if** `isConstrained` **then**
22:        constrained $\leftarrow$ constrained $+$ detokenize($t_i$)
23:        **if** constrained.endswith($S_2$) **then**
24:           `pointer` $\leftarrow$ len(tokens)
25: **return** detokenize(tokens)

---

If $S_1$ is not present, the next token is computed directly without any constraints applied. Additionally, if the current constrained window ends with $S_2$, the pointer is updated to the length of the current token sequence, effectively switching back to unconstrained generation until $S_1$ is generated again. Figure 2 further illustrates LLM generation with CRANE. The underlined portion of the LLM generation represents `currGen`, and the current constrained window is highlighted in yellow.

## 5. Evaluation

In this section, we evaluate CRANE on a math reasoning task (GSM-Symbolic (Mirzadeh et al., 2024)) and a logical reasoning task (FOLIO (Han et al., 2024)) and demonstrate significant improvement over both unconstrained and SOTA constrained generation baselines.

**Experimental Setup.** We run experiments on a 48-core Intel Xeon Silver 4214R CPU with 2 NVidia RTX A5000 GPUs. CRANE is implemented using PyTorch (Paszke et al., 2019) and the HuggingFace transformers library (Wolf et al., 2020). Our primary baseline for unconstrained generation is Chain-of-Thought (CoT) Prompting (Wei et al.,

2022), which enables LLMs to decompose and reason about a problem through a series of intermediate steps before outputting the final answer. Furthermore, we run constrained semantic generation for GSM-Symbolic (Mirzadeh et al., 2024) with the ITERGEN library (Ugare et al., 2024a) and use the SYNCODE framework for FOLIO (Han et al., 2024) evaluation. In all experiments, CRANE is initialized with the same constrained decoders and uses the same constraints as the constrained generation baselines.

**GSM-Symbolic:** We first evaluate CRANE on GSM-Symbolic (Mirzadeh et al., 2024), a dataset consisting of math word problems designed to assess LLMs' mathematical reasoning skills. In the word problems, names and numerical values are replaced with symbolic variables, and the LLMs are tasked with generating correct symbolic expression solutions (see Appendix D.1 for examples). To evaluate correctness, we extract the final expressions from the LLM generations and verify if they are functionally equivalent to the ground truth expressions with the Z3 solver (De Moura & Bjørner, 2008).

We compare CRANE against three baselines: (1) unconstrained generation without chain-of-thought prompting, (2) unconstrained generation with CoT, and (3) constrained generation. We use ITERGEN for the constrained generation baseline and also initialize CRANE with ITERGEN. For ITERGEN and CRANE, we enforce syntactic constraints via the context-free grammar provided in Appendix D.5.1 and apply the semantic constraint ensuring that generated expressions contain only valid problem-defined variables. Since ITERGEN uses selective rejection sampling to enforce semantic constraints, we also include comparisong against unconstrained generation with sampling in Table 4 in the Appendix. For CRANE, we use << and >> for the delimeters $S_1$ and $S_2$, respectively. We evaluate Qwen2.5-1.5B-Instruct (Qwen, 2024), Qwen2.5-Math-7B-Instruct (Qwen, 2024), Qwen2.5-Coder-7B-Instruct (Qwen, 2024),Llama-3.1-8B-Instruct (Llama, 2024), DeepSeek-R1-Distill-Qwen-7B (DeepSeek-AI et al., 2025), DeepSeek-R1-Distill-Llama-8B (DeepSeek-AI et al., 2025), Qwen2.5-Coder-14B-Instruct (Qwen, 2024), DeepSeek-R1-Distill-Qwen-14B (DeepSeek-AI et al., 2025), and QwQ-32B (Team, 2025). We use greedy decoding with a maximum new token limit of 600 and prompt the LLMs with the 8-shot examples from GSM-Symbolic (Mirzadeh et al., 2024) (the prompts can be found in Appendix D.1).

Table 1 compares the performance of CRANE with the baseline methods. The Accuracy (%) column reports the percentage of functionally correct LLM-generated expressions, Parse (%) indicates the percentage of syntactically valid expressions (i.e., expressions without invalid operations), and Tokens provides the average number of tokens generated.

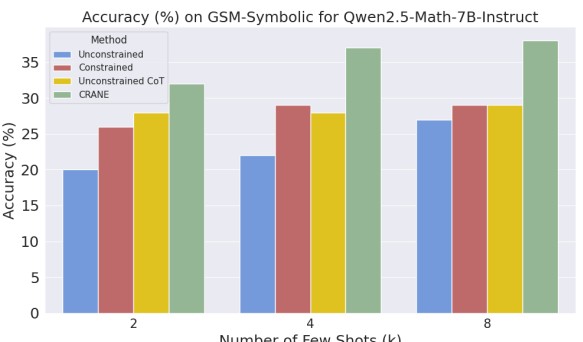

*Figure 3.* Accuracy (%) of Qwen2.5-Math-7B-Instruct By Method and Number of Shots on GSM-Symbolic

As shown in the table, CRANE consistently improves functional correctness across all evaluated models. For example, with the Qwen2.5-Math-7B-Instruct model, CRANE achieves 38% accuracy, outperforming both constrained generation and unconstrained generation with CoT, which achieves 29% accuracy. Moreover, CRANE significantly enhances the syntactic correctness of generated expressions compared to unconstrained generation. Notably, none of the expressions generated using CRANE contain syntax errors, whereas 10% of the expressions from unconstrained generation with CoT do. Although, for several instances, CRANE produces slightly more syntax errors than a purely constrained generation, it offers a substantial improvement in functional correctness over this baseline.

**Ablation Study on Few-shot examples:** We evaluate CRANE and baselines on varying numbers of few-shot examples in the prompt and display the results for Qwen2.5-Math-7B-Instruct in Figure 3. Results for all models are presented in Table 3 in the Appendix. CRANE consistently achieves higher accuracy on GSM-Symbolic than the baselines for all evaluated numbers of few-shot examples.

**FOLIO:** We further evaluate CRANE on the validation split of FOLIO dataset, which comprises 203 expert-written natural language reasoning instances and corresponding first-order logic (FOL) annotations. We evaluate the ability of LLMs to correctly translate the natural language reasoning instances into FOL formulas and leverage Prover9 (Mc-Cune, 2005–2010) a FOL solver to verify the correctness of the LLM-generated FOL formulas.

We compare CRANE against grammar-constrained generation with SYNCODE using the Prover9 grammar (Appendix D.5.2). The Prover9 grammar divides FOL formulas into Predicates, Premises, and Conclusions and allows intermediate reasoning in comments (an example can be found in Appendix D.2). We also compare CRANE against unconstrained generation with CoT. For all approaches and models, we run greedy decoding with a maximum new to-

kens limit of 800 and use 2 few-shot examples in the prompt. We also compare CRANE against unconstrained CoT with temperature sampling in Table 5 in the Appendix.

Appendix table 2 presents the results of our experiment. The Accuracy (%) column in the table reports the percentage of functionally correct FOL translations while the Compiles (%) column reports the percentage of FOL formulas extracted from LLM output that are syntactically valid and compile into a Prover9 program. CRANE outperforms the unconstrained and constrained generation baselines for all models evaluated.

**Limitation:** Our work has the following limitations. First, Proposition 3.1 only demonstrates a reduction in expressivity when the language $L(G_c)$ is finite. This leaves open the question of whether Proposition 3.1 can be extended to grammars $G$ where $L(G)$ is infinite. Second, CRANE for constrained decoding relies on existing tools (Ugare et al., 2024b) that require access to output logits, rendering CRANE inapplicable to models that do not expose logits.

## 6. Related Works

**Constrained LLM Decoding:** Recent works have introduced techniques to enforce LLM generations to adhere to a context-free grammar using constrained decoding (Ugare et al., 2024c; Willard & Louf, 2023; Beurer-Kellner et al., 2024; Melcer et al., 2024a). Additionally, Poesia et al. (2022); Ugare et al. (2024a) have extended grammar-guided generation to incorporate task-specific semantic constraints. These approaches demonstrate that constrained decoding can improve the syntactic and semantic quality of LLM outputs for various structured generation tasks.

More recently, Tam et al. (2024) demonstrated that constrained structured generation can negatively impact the quality of generated outputs. Similarly, Park et al. (2024) showed that greedily masking out tokens that do not lead to a valid string during next-token prediction can distort the output distribution, causing it to deviate from the true distribution of all grammatically valid outputs of $\mathcal{L}$ for a given input. To mitigate the distortion introduced by the greedy masking approach, these "grammar aligned" methods (Park et al., 2024; Melcer et al., 2024b) use a trie to track previous generations, reducing generation divergence iteratively. However, they are computationally expensive and require a large no. of resamplings per prompt to converge.

In contrast, our work focuses on the fundamental question of the theoretical expressivity of any constant layered constrained LLM, even under an ideal constrained decoding algorithm, and uses the insights to propose a practical solution. We propose an adaptive constrained decoding approach that can support various constrained decoding methods, including grammar-aligned techniques while preserv-

ing the LLM's expressivity by reasoning chains.

**LLM Expressivity:** (Strobl et al., 2024) provides a detailed survey of existing results from the perspective of formal language theory and complexity classes. A series of existing works (Hahn, 2020; Hao et al., 2022; Merrill et al., 2022; Merrill & Sabharwal, 2023) establish that, under suitable assumptions, a single autoregressive step on an input of any length for a constant-depth LLM can be represented as a constant-depth Boolean circuit. (Merrill & Sabharwal, 2024; Li et al., 2024) show that the expressivity of LLMs significantly improves under popular reasoning approaches like Chain of Thought (CoT) (Wei et al., 2022), where LLMs take intermediate steps before generating the final answer. To the best of our knowledge, there is no prior work on LLM expressivity under grammar constraints.

## 7. Conclusion

In conclusion, tasks requiring both syntactic and semantic correctness, such as code generation and symbolic math reasoning, benefit significantly from constrained decoding strategies. However, strict enforcement of constraints can hinder LLM reasoning capabilities. Theoretically, we demonstrate why restrictive grammars diminish reasoning and show that augmenting grammars with carefully designed rules preserves reasoning while maintaining correctness. Building on these insights, our proposed reasoning-augmented constrained decoding algorithm, CRANE, achieves state-of-the-art performance, with up to 10% improvement on symbolic reasoning benchmarks such as GSM-symbolic and FOLIO, effectively balancing the strengths of constrained and unconstrained generation.

Table 1. Comparison of CRANE and baselines with different models on GSM-Symbolic.

| Model | Method | Acc. (%) | Parse (%) | Tokens |
|---|---|---|---|---|
| Qwen2.5-1.5B-Instruct | Unconstrained w/o CoT | 21 | 97 | 23.34 |
| | Constrained | 22 | 97 | 25.29 |
| | Unconstrained CoT | 26 | 90 | 128.97 |
| | **CRANE** | **31** | 100 | 131.3 |
| Qwen2.5-Coder-7B-Instruct | Unconstrained w/o CoT | 36 | 94 | 17.92 |
| | Constrained | 35 | 99 | 25.28 |
| | Unconstrained CoT | 37 | 88 | 138.38 |
| | **CRANE** | **39** | 94 | 155.32 |
| Qwen2.5-Math-7B-Instruct | Unconstrained w/o CoT | 27 | 89 | 25.7 |
| | Constrained | 29 | 99 | 26.81 |
| | Unconstrained CoT | 29 | 82 | 155.26 |
| | **CRANE** | **38** | 94 | 158.86 |
| Llama-3.1-8B-Instruct | Unconstrained w/o CoT | 21 | 73 | 128.38 |
| | Constrained | 26 | 98 | 35.97 |
| | Unconstrained CoT | 30 | 95 | 163.55 |
| | **CRANE** | **33** | 95 | 170.22 |
| DeepSeek-R1-Distill-Qwen-7B | Unconstrained w/o CoT | 18 | 89 | 21.64 |
| | Constrained | 20 | 99 | 17.21 |
| | Unconstrained CoT | 24 | 89 | 212.24 |
| | **CRANE** | **29** | 92 | 235.78 |
| DeepSeek-R1-Distill-Llama-8B | Unconstrained w/o CoT | 12 | 77 | 29.2 |
| | Constrained | 13 | 96 | 16.89 |
| | Unconstrained CoT | 21 | 87 | 250.83 |
| | **CRANE** | **31** | 92 | 268.82 |
| Qwen2.5-Coder-14B-Instruct | Unconstrained w/o CoT | 42 | 93 | 18.52 |
| | Constrained | 42 | 96 | 25.62 |
| | Unconstrained CoT | 42 | 95 | 157.71 |
| | **CRANE** | **45** | 95 | 158.54 |
| DeepSeek-R1-Distill-Qwen-14B | Unconstrained w/o CoT | 29 | 82 | 20.9 |
| | Constrained | 30 | 91 | 30.48 |
| | Unconstrained CoT | 32 | 87 | 233.42 |
| | **CRANE** | **38** | 93 | 244.98 |
| QwQ-32B | Unconstrained w/o CoT | 37 | 80 | 54.93 |
| | Constrained | 38 | 91 | 34.2 |
| | Unconstrained CoT | 43 | 87 | 222.62 |
| | **CRANE** | **46** | 88 | 237.98 |

## Impact Statement

This paper introduces research aimed at advancing the field of Machine Learning. We do not identify any specific societal consequences of our work that need to be explicitly emphasized here.

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

# A. Turing Machine Computation

A Turing machine processes an input string $\boldsymbol{x} \in \Sigma^*$. Its configuration consists of a finite state set $Q$, an input tape $c_0$, $k$ work tapes $c_1, \ldots, c_k$, and an output tape $c_{k+1}$. Additionally, each tape $\tau$ has an associated head position $h_\tau$.

Initially, the machine starts in the initial state $q_0 \in Q$ with the input tape $c_0^0$ containing $\boldsymbol{x}$, positioned at index $0$, and surrounded by infinite blank symbols ($b$). The head on the input tape is set to $h_0^0 = 0$, while all other tapes contain only blank symbols $b$s and have their heads positioned at $0$.

At each time step $i$, if $q_i \notin F$ ($F$ is a set of halting states), the configuration updates recursively by computing:

$$\langle q_{i+1}, \gamma_1^i, \ldots, \gamma_{k+1}^i, d_0^i, \ldots, d_{k+1}^i \rangle = \delta(q_i, c_0^i[h_0^i], \ldots, c_{k+1}^i[h_{k+1}^i])$$

where $\delta$ is the transition function. The machine updates each tape $\tau$ by setting $c_\tau^{i+1}[h_\tau^i] = \gamma_\tau^i$, leaving all other tape cells unchanged. The head position for each tape is updated as $h_\tau^{i+1} = h_\tau^i + d_\tau^i$. If $q_i \in F$, the Turing machine halts and outputs the sequence of tokens on the output tape, starting from the current head position and continuing up to (but not including) the first blank symbol ($b$). A Turing machine can also function as a language recognizer by setting the input alphabet $\Sigma = \{0, 1\}$ and interpreting the first output token as either $0$ or $1$.

# B. Threshold Circuit Class

$TC^0$ is a class of computational problems that can be recognized by constant-depth, polynomial-size circuits composed of threshold gates. A threshold gate, such as $\theta_{\leq k}$, outputs 1 if the sum of its input bits is at most $k$, while $\theta_{>k}$ outputs 1 if the sum is at least $k$. These circuits also include standard logic gates like $\wedge$, $\vee$, and $\neg$ as special cases of threshold functions. Since $TC^0$ circuits can simulate $AC^0$ circuits ( a polysize, constant-depth $\{\wedge, \vee, \neg\}$-circuit family), they are at least as powerful as $AC^0$ in the computational hierarchy. The circuit families we have defined above are non-uniform, meaning that there is no requirement for the circuits processing different input sizes to be related in any way. In degenerate cases, non-uniform circuit families can solve undecidable problems making them an unrealizable model of computation (Arora & Barak, 2009). Intuitively, a uniform circuit family requires that the circuits for different input sizes must be "somewhat similar" to each other. This concept is formalized by stating that there exists a resource-constrained Turing machine that, given the input $1^n$, can generate a serialization of the corresponding circuit $C_n$ for that input size. Specifically, a logspace uniform $TC^0$ family can be constructed by a logspace-bounded Turing machine from the string $1^n$.

# C. Proofs

**Lemma C.1** (Constant depth circuit for $\mathcal{L}_f$). *For any log-precision constant layer transformer-based LLM $\mathcal{L}$ with finite vocabulary $V$, a single deterministic auto-regressive step $\mathcal{L}_f(x)$ operating on any input of size $n \in \mathbb{N}$ with $\boldsymbol{x} \in V^n$ can be simulated by a logspace-uniform threshold circuit family of depth $C$ where $C$ is constant.*

*Proof.* The construction is from Theorem 2 in (Merrill & Sabharwal, 2023). $\square$

**Proposition 3.1.** *For any log-precision LLM $\mathcal{L}$ with constant layers there exists a logspace-uniform thershold circuit $Th_n$ such that $\mathcal{L}_{G_c}(\boldsymbol{x}) = Th_n(\boldsymbol{x})$ holds for all inputs $\boldsymbol{x}$ with size $|\boldsymbol{x}| = n$ and $n \in \mathbb{N}$.*

*Proof.* The language $L(G_c)$ is finite; therefore, for any string $\boldsymbol{s} \in L(G_c)$, the length satisfies $|\boldsymbol{s}| \leq N$, where $N$ is a constant. Consequently, for any input $\boldsymbol{x}$, the output $\boldsymbol{y}_G = \mathcal{L}_G(\boldsymbol{x})$ has a constant length, i.e., $|\boldsymbol{y}_G| \leq N$. The number of autoregressive steps is also bounded by $N$.

From Lemma C.1, each unconstrained autoregressive computation $\mathcal{L}_f(\boldsymbol{x})$ can be simulated by a constant-depth threshold circuit $C$. This implies that $\mathcal{L}_f(\boldsymbol{x}, G_c)$ can also be simulated by a constant-depth threshold circuit since it only involves an additional multiplication by a constant-sized precomputed Boolean mask $\{0, 1\}^{|V|}$ (see Section 2).

Given that the number of autoregressive steps is a constant $N$, and each step can be simulated by a constant-depth circuit $C$, we can simulate all $N$ steps using a depth $N \times C$ circuit by stacking the circuits for each step sequentially. For uniformity, we are just stacking together a constant number of constant depth circuits we can do it in a log-space bounded Turning machine $M$.

Note that this proof holds only because $L(G_c)$ allows only constant-size strings in the output. $\square$

**Lemma 3.2.** *For any Turing machine $M$ with tape alphabet $\Gamma$, there exists a constant depth LLM $\mathcal{L}_M$ with finite vocabulary $\Gamma \subseteq V_M$ and log-precision that can simulate $t(n)$ steps of $M$ with $t(n)$ autoregressive steps.*

*Proof.* The construction follows from Theorem 2 (Merrill & Sabharwal, 2024). □

In this construction, the deterministic Turing machine run captured by a sequence of $\overline{\gamma_1}, \ldots, \overline{\gamma_{t(n)}}$ capturing the state entered, tokens written, and directions moved after each token before generating the output $M(\boldsymbol{x})$. Then on any input the $\boldsymbol{x}$ the output $\mathcal{L}_M(\boldsymbol{x}) = \overline{\gamma_1}, \cdots, \overline{\gamma_{t(n)}} \cdot M(\boldsymbol{x})$ (assuming $M$ halts within on $\boldsymbol{x}$ within $t(n)$ steps where $n = |\boldsymbol{x}|$ and $t(n)$ is a polynomial over $n$).

**Proposition 3.3.** *For any Turing machine $M$ with tape alphabet $\Gamma$, there exists a constant depth LLM $\mathcal{L}_M$ with finite vocabulary $\Gamma \subseteq V_M$ and log precision such that for any input $\boldsymbol{x}$ with $|\boldsymbol{x}| = n$, $\mathcal{L}_{M,G_a}(\boldsymbol{x}) = \boldsymbol{r} \cdot M(\boldsymbol{x})$ with $r \in V_M^*$ assuming $M$ halts on $\boldsymbol{x}$ in $t(n)$ steps.*

*Proof.* $\mathcal{L}_M(\boldsymbol{x})) = \overline{\gamma_1} \cdots \overline{\gamma_{t(n)}} \cdot M(\boldsymbol{x})$. We show that $\mathcal{L}_M(\boldsymbol{x}) \in L(G_a)$. $G_a \rightarrow R_M G$. Since, $G$ is output grammar of $M$ then $M(\boldsymbol{x}) \in L(G)$. For all $1 \leq i \leq t(n)$ $\overline{\gamma_i} \in \overline{\Gamma}$. Then, $\overline{\gamma_1} \cdots \overline{\gamma_{t(n)}} \in \overline{\Gamma}^* \subseteq L(R_M)$.

Then $\mathcal{L}_M(\boldsymbol{x}) \in L(G_a)$ then under constrained decoding the output $\mathcal{L}_M(\boldsymbol{x})$ remains unchanged and $\mathcal{L}_M(\boldsymbol{x}) = \mathcal{L}_{M,G_a}(\boldsymbol{x}) = \boldsymbol{r} \cdot M(\boldsymbol{x})$ where $\boldsymbol{r} = \overline{\gamma_1} \cdots \overline{\gamma_{t(n)}}$. □

# D. FOLIO Results

*Table 2.* Comparison of CRANE and baselines with various models on FOLIO.

| Model | Method | Acc. (%) | Compiles (%) | Tokens |
|---|---|---|---|---|
| Qwen2.5-Math-7B-Instruct | Unconstrained CoT | 18.72 | 54.19 | 629.59 |
| | Constrained | 28.08 | 76.85 | 679.44 |
| | **CRANE** | **31.03** | 75.86 | 690.17 |
| Qwen2.5-7B-Instruct | Unconstrained CoT | 36.95 | 70.94 | 350.64 |
| | Constrained | 37.44 | 87.68 | 775.62 |
| | **CRANE** | **42.36** | 87.68 | 726.88 |
| Llama-3.1-8B-Instruct | Unconstrained CoT | 32.02 | 57.14 | 371.52 |
| | Constrained | 39.41 | 86.21 | 549.75 |
| | **CRANE** | **46.31** | 85.71 | 449.77 |

## D.1. GSM-Symbolic Examples and Prompt

**GSM-Symbolic Problem Solution Examples:**

```
1  Question: A fog bank rolls in from the ocean to cover a city. It takes {t} minutes to cover every {d}
       miles of the city. If the city is {y} miles across from oceanfront to the opposite inland edge,
       how many minutes will it take for the fog bank to cover the whole city?
2
3  Answer: y//d*t
4
5  Question: {name} makes {drink} using teaspoons of sugar and cups of water in the ratio of {m}:{n}. If
       she used a total of {x} teaspoons of sugar and cups of water, calculate the number of
       teaspoonfuls of sugar she used.
6
7  Answer: ((m*x)//(m+n))
```

*Listing 1.* Problem Solution Examples for GSM-Symbolic

**GSM-Symbolic Prompt:**

```
1  You are an expert in solving grade school math tasks. You will be presented with a grade-school math
       word problem with symbolic variables and be asked to solve it.
2
3  Before answering you should reason about the problem (using the <reasoning> field in the response
       described below). Intermediate symbolic expressions generated during reasoning should be wrapped
       in << >>.
4
5  Then, output the symbolic expression wrapped in << >> that answers the question. The expressions must
       use numbers as well as the variables defined in the question. You are only allowed to use the
       following operations: +, -, /, //, %, (), and int().
6
7  You will always respond in the format described below:
8  Let's think step by step. <reasoning> The final answer is <<symbolic expression>>
9
10 There are {t} trees in the {g}. {g} workers will plant trees in the {g} today. After they are done,
       there will be {tf} trees. How many trees did the {g} workers plant today?
11
12 Let's think step by step. Initially, there are {t} trees. After planting, there are {tf} trees. The
       number of trees planted is <<tf - t>>. The final answer is <<tf - t>>.
13
14 If there are {c} cars in the parking lot and {nc} more cars arrive, how many cars are in the parking
       lot?
15
16 Let's think step by step. Initially, there are {c} cars. {nc} more cars arrive, so the total becomes
       <<c + nc>>. The final answer is <<c + nc>>.
17
18 {p1} had {ch1} {o1} and {p2} had {ch2} {o1}. If they ate {a} {o1}, how many pieces do they have left
       in total?
19
20 Let's think step by step. Initially, {p1} had {ch1} {o1}, and {p2} had {ch2} {o1}, making a total of
       <<ch1 + ch2>>. After eating {a} {o1}, the remaining total is <<ch1 + ch2 - a>>. The final answer
       is <<ch1 + ch2 - a>>.
21
22 {p1} had {l1} {o1}. {p1} gave {g} {o1} to {p2}. How many {o1} does {p1} have left?
23
```

```
24  Let's think step by step. {p1} started with {l1} {o1}. After giving {g} {o1} to {p2}, {p1} has <<l1 -
        g>> {o1} left. The final answer is <<l1 - g>>.
25
26  {p1} has {t} {o1}. For Christmas, {p1} got {tm} {o1} from {p2} and {td} {o1} from {p3}. How many {o1}
        does {p1} have now?
27
28  Let's think step by step. {p1} started with {t} {o1}. {p1} received {tm} {o1} from {p2} and {td} {o1}
        from {p3}. The total is <<t + tm + td>>. The final answer is <<t + tm + td>>.
29
30  There were {c} {o1} in the server room. {nc} more {o1} were installed each day, from {d1} to {d2}. How
         many {o1} are now in the server room?
31
32  Let's think step by step. Initially, there were {c} {o1}. {nc} {o1} were added each day for <<d2 - d1
        + 1>> days, which is <<nc * (d2 - d1 + 1)>>. The total is <<c + nc * (d2 - d1 + 1)>>. The final
        answer is <<c + nc * (d2 - d1 + 1)>>.
33
34  {p1} had {gb1} {o1}. On {day1}, {p1} lost {l1} {o1}. On {day2}, {p1} lost {l2} more. How many {o1}
        does {p1} have at the end of {day2}?
35
36  Let's think step by step. Initially, {p1} had {gb1} {o1}. After losing {l1} {o1} on {day1}, {p1} had
        <<gb1 - l1>>. After losing {l2} {o1} on {day2}, the total is <<gb1 - l1 - l2>>. The final answer
        is <<gb1 - l1 - l2>>.
37
38  {p1} has ${m}. {p1} bought {q} {o1} for ${p} each. How much money does {p1} have left?
39
40  Let's think step by step. Initially, {p1} had ${m}. {p1} spent <<q * p>> on {q} {o1}. The remaining
        money is <<m - q * p>>. The final answer is <<m - q * p>>.
41
42  {question}
```

*Listing 2.* CoT Prompt Template For GSM-Symbolic Evaluation

```
1   You are an expert in solving grade school math tasks. You will be presented with a grade-school math
        word problem with symbolic variables and be asked to solve it.
2
3   Only output the symbolic expression wrapped in << >> that answers the question. The expression must
        use numbers as well as the variables defined in the question. You are only allowed to use the
        following operations: +, -, /, //, %, (), and int().
4
5   You will always respond in the format described below:
6   <<symbolic expression>>
7
8   There are {t} trees in the {g}. {g} workers will plant trees in the {g} today. After they are done,
        there will be {tf} trees. How many trees did the {g} workers plant today?
9
10  <<tf - t>>
11
12  If there are {c} cars in the parking lot and {nc} more cars arrive, how many cars are in the parking
        lot?
13
14  <<c + nc>>
15
16  {p1} had {ch1} {o1} and {p2} had {ch2} {o1}. If they ate {a} {o1}, how many pieces do they have left
        in total?
17
18  <<ch1 + ch2 - a>>
19
20  {p1} had {l1} {o1}. {p1} gave {g} {o1} to {p2}. How many {o1} does {p1} have left?
21
22  <<l1 - g>>
23
24  {p1} has {t} {o1}. For Christmas, {p1} got {tm} {o1} from {p2} and {td} {o1} from {p3}. How many {o1}
        does {p1} have now?
25
26  <<t + tm + td>>
27
28  There were {c} {o1} in the {loc}. {nc} more {o1} were installed each day, from {d1} to {d2}. How many
        {o1} are now in the {loc}?
29
30  <<c + nc * (d2 - d1 + 1)>>
31
32  {p1} had {gb1} {o1}. On {day1}, {p1} lost {l1} {o1}. On {day2}, {p1} lost {l2} more. How many {o1}
        does {p1} have at the end of {day2}?
33
34  <<gb1 - l1 - l2>>
35
36  {p1} has ${m}. {p1} bought {q} {o1} for ${p} each. How much money does {p1} have left?
37
38  <<m - q * p>>
```

```
39
40  {question}
```

*Listing 3.* Prompt Template For GSM-Symbolic Evaluation Without CoT

## D.2. FOLIO Examples and Prompt

### FOLIO Problem Solution Examples:

```
1   Question:
2   People in this club who perform in school talent shows often attend and are very engaged with school
         events.
3   People in this club either perform in school talent shows often or are inactive and disinterested
         community members.
4   People in this club who chaperone high school dances are not students who attend the school.
5   All people in this club who are inactive and disinterested members of their community chaperone high
         school dances.
6   All young children and teenagers in this club who wish to further their academic careers and
         educational opportunities are students who attend the school.
7   Bonnie is in this club and she either both attends and is very engaged with school events and is a
         student who attends the school or is not someone who both attends and is very engaged with school
          events and is not a student who attends the school.
8   Based on the above information, is the following statement true, false, or uncertain? Bonnie performs
         in school talent shows often.
9   ###
10
11  FOL Solution:
12  Predicates:
13  InClub(x) ::: x is a member of the club.
14  Perform(x) ::: x performs in school talent shows.
15  Attend(x) ::: x attends school events.
16  Engaged(x) ::: x is very engaged with school events.
17  Inactive(x) ::: x is an inactive and disinterested community member.
18  Chaperone(x) ::: x chaperones high school dances.
19  Student(x) ::: x is a student who attends the school.
20  Wish(x) ::: x wishes to further their academic careers and educational opportunities.
21  Premises:
22  {forall} x (InClub(x) {and} Attend(x) {and} Engaged(x) {implies} Attend(x)) ::: People in this club
         who perform in school talent shows often attend and are very engaged with school events.
23  {forall} x (InClub(x) {implies} (Perform(x) {xor} Inactive(x))) ::: People in this club either perform
          in school talent shows often or are inactive and disinterested community members.
24  {forall} x (InClub(x) {and} Chaperone(x) {implies} {not}Student(x)) ::: People in this club who
         chaperone high school dances are not students who attend the school.
25  {forall} x (InClub(x) {and} Inactive(x) {implies} Chaperone(x)) ::: All people in this club who are
         inactive and disinterested members of their community chaperone high school dances.
26  {forall} x (InClub(x) {and} (Young(x) {or} Teenager(x)) {and} Wish(x) {implies} Student(x)) ::: All
         young children and teenagers in this club who wish to further their academic careers and
         educational opportunities are students who attend the school.
27  {forall} x (InClub(x) {implies} (Attend(x) {and} Engaged(x)) {xor} {not}(Attend(x) {and} Engaged(x)) {
         and} {not}Student(x) {xor} Student(x)) ::: Bonnie is in this club and she either both attends and
          is very engaged with school events and is a student who attends the school or is not someone who
          both attends and is very engaged with school events and is not a student who attends the school.
28  Conclusion:
29  InClub(bonnie) {and} Perform(bonnie) ::: Bonnie performs in school talent shows often.
30
31  Answer: Uncertain
```

*Listing 4.* Problem Solution Examples for FOLIO

### FOLIO Prompt:

```
1   Given a problem description and a question. The task is to parse the problem and the question into
         first-order logic formulas.
2   The grammar of the first-order logic formula is defined as follows:
3   1) logical conjunction of expr1 and expr2: expr1 {and} expr2
4   2) logical disjunction of expr1 and expr2: expr1 {or} expr2
5   3) logical exclusive disjunction of expr1 and expr2: expr1 {xor} expr2
6   4) logical negation of expr1: {not}expr1
7   5) expr1 implies expr2: expr1 {implies} expr2
8   6) expr1 if and only if expr2: expr1 {iff} expr2
9   7) logical universal quantification: {forall} x
10  8) logical existential quantification: {exists} x. These are the ONLY operations in the grammar.
11  ------
12
13  Answer the question EXACTLY like the examples.
14
```

```
15  Problem:
16  All people who regularly drink coffee are dependent on caffeine. People either regularly drink coffee
        or joke about being addicted to caffeine. No one who jokes about being addicted to caffeine is
        unaware that caffeine is a drug. Rina is either a student and unaware that caffeine is a drug, or
         neither a student nor unaware that caffeine is a drug. If Rina is not a person dependent on
        caffeine and a student, then Rina is either a person dependent on caffeine and a student, or
        neither a person dependent on caffeine nor a student.
17  Question:
18  Based on the above information, is the following statement true, false, or uncertain? Rina is either a
         person who jokes about being addicted to caffeine or is unaware that caffeine is a drug.
19  ###
20
21  We take three steps: first, we define the necessary predicates and premises, and finally, we encode
        the question 'Rina is either a person who jokes about being addicted to caffeine or is unaware
        that caffeine is a drug.' in the conclusion. Now, we will write only the logic program, nothing
        else.
22  Predicates:
23  Dependent(x) ::: x is a person dependent on caffeine.
24  Drinks(x) ::: x regularly drinks coffee.
25  Jokes(x) ::: x jokes about being addicted to caffeine.
26  Unaware(x) ::: x is unaware that caffeine is a drug.
27  Student(x) ::: x is a student.
28  Premises:
29  {forall} x (Drinks(x) {implies} Dependent(x)) ::: All people who regularly drink coffee are dependent
        on caffeine.
30  {forall} x (Drinks(x) {xor} Jokes(x)) ::: People either regularly drink coffee or joke about being
        addicted to caffeine.
31  {forall} x (Jokes(x) {implies} {not}Unaware(x)) ::: No one who jokes about being addicted to caffeine
        is unaware that caffeine is a drug.
32  (Student(rina) {and} Unaware(rina)) {xor} {not}(Student(rina) {or} Unaware(rina)) ::: Rina is either a
         student and unaware that caffeine is a drug, or neither a student nor unaware that caffeine is a
         drug.
33  Conclusion:
34  Jokes(rina) {xor} Unaware(rina) ::: Rina is either a person who jokes about being addicted to caffeine
         or is unaware that caffeine is a drug.
35  ------
36
37  Problem:
38  Miroslav Venhoda was a Czech choral conductor who specialized in the performance of Renaissance and
        Baroque music. Any choral conductor is a musician. Some musicians love music. Miroslav Venhoda
        published a book in 1946 called Method of Studying Gregorian Chant.
39  Question:
40  Based on the above information, is the following statement true, false, or uncertain? Miroslav Venhoda
         loved music.
41  ###
42
43  We take three steps: first, we define the necessary predicates and premises, and finally, we encode
        the question 'Miroslav Venhoda loved music.' in the conclusion. Now, we will write only the logic
         program, nothing else.
44  Predicates:
45  Czech(x) ::: x is a Czech person.
46  ChoralConductor(x) ::: x is a choral conductor.
47  Musician(x) ::: x is a musician.
48  Love(x, y) ::: x loves y.
49  Author(x, y) ::: x is the author of y.
50  Book(x) ::: x is a book.
51  Publish(x, y) ::: x is published in year y.
52  Specialize(x, y) ::: x specializes in y.
53  Premises:
54  Czech(miroslav) {and} ChoralConductor(miroslav) {and} Specialize(miroslav, renaissance) {and}
        Specialize(miroslav, baroque) ::: Miroslav Venhoda was a Czech choral conductor who specialized
        in the performance of Renaissance and Baroque music.
55  {forall} x (ChoralConductor(x) {implies} Musician(x)) ::: Any choral conductor is a musician.
56  {exists} x (Musician(x) {and} Love(x, music)) ::: Some musicians love music.
57  Book(methodOfStudyingGregorianChant) {and} Author(miroslav, methodOfStudyingGregorianChant) {and}
        Publish(methodOfStudyingGregorianChant, year1946) ::: Miroslav Venhoda published a book in 1946
        called Method of Studying Gregorian Chant.
58  Conclusion:
59  Love(miroslav, music) ::: Miroslav Venhoda loved music.
60  ------
61
62  {question}
```

*Listing 5.* Prompt Template Used For FOLIO Evaluation

### D.3. Case Study For GSM-Symbolic

```
1  Question: {name} hires a {vehicle} from {start_hour}  to {end_hour}. He gets {free_hours} hours free.
       The first paid hour is ${first_hour_cost} and each hour after that is {multiplier} the cost. How
       much did he pay?
2
3  Constrained: <<(int(end_hour - start_hour) - free_hours) * first_hour_cost + free_hours *
       first_hour_cost + (int(end_hour - start_hour) - free_hours - 1) * multiplier * first_hour_cost>>
4
5  Unconstrained With CoT:  Let's think step by step. The total time hired is from {start_hour} to {
       end_hour}, which is <<int((end_hour - start_hour).total_seconds() / 3600)>> hours. Subtracting
       the {free_hours} free hours, the paid hours are <<int((end_hour - start_hour).total_seconds() /
       3600) - free_hours>>. The final answer is <<first_hour_cost + (int((end_hour - start_hour).
       total_seconds() / 3600) - free_hours - 1) * multiplier * first_hour_cost>>.
6
7  CRANE: Let's think step by step. The total time hired is from {start_hour} to {end_hour}, totaling <<
       int(end_hour - start_hour)>> hours. Subtracting {free_hours} free hours, the paid hours are <<int
       (end_hour - start_hour) - free_hours>>. The first hour costs {first_hour_cost}, and each
       additional hour costs {multiplier} times that. The final answer is <<first_hour_cost + (int(
       end_hour - start_hour) - free_hours - 1) * multiplier * first_hour_cost>>.
```

*Listing 6.* Case Study for GSM-Symbolic

CRANE effectively alternates between constrained and unconstrained generation to produce intermediate expressions, the final answer, and to maintain the reasoning capabilities of the LLM. In contrast, unconstrained generation with CoT results in a syntactically incorrect expression, while constrained generation produces a syntactically valid but incorrect expression.

### D.4. Sampling Ablation for GSM-Symbolic

In our GSM-Symbolic case study, we use IterGen as the constrained generation baseline and initialize CRANE with IterGen. Both IterGen and CRANE employ selective rejection sampling to filter tokens that do not satisfy semantic constraints. For comparison, we also run unconstrained generation using temperature sampling and evaluate its performance against CRANE. Specifically, for Qwen2.5-1.5B-Instruct and Llama-3.1-8B-Instruct, we generate three samples with unconstrained generation at a temperature of $t = 0.7$ and compute pass@1/2/3 metrics.

As shown in Table 4, CRANE with greedy decoding achieves higher accuracy than pass@1/2/3 for unconstrained generation with Chain-of-Thought (CoT) and temperature sampling on Qwen2.5-1.5B-Instruct. Although, for Llama-3.1-8B-Instruct, unconstrained generation with CoT and temperature sampling achieves a pass@3 accuracy of 35%—2% higher than CRANE—it generates approximately 4 times as many tokens as CRANE.

### D.5. Grammars

#### D.5.1. GSM-SYMBOLIC GRAMMAR

```
1  start: space? "<" "<" space? expr space? ">" ">" space?
2
3  expr: expr space? "+" space? term
4      | expr space? "-" space? term
5      | term
6
7  term: term space? "*" space? factor
8      | term space? "/" space? factor
9      | term space? "//" space? factor
10     | term space? "%" space? factor
11     | factor space?
12
13 factor: "-" space? factor
14       | TYPE "(" space? expr space? ")"
15       | primary space?
16
17 primary: NUMBER
18        | VARIABLE
19        | "(" space? expr space? ")"
20
21 TYPE.4: "int"
22
23 space: " "
24
25 %import common.CNAME -> VARIABLE
26 %import common.NUMBER
```

*Listing 7.* GSM-Symbolic Grammar

### D.5.2. PROVER9 GRAMMAR

```
1   start: predicate_section premise_section conclusion_section
2
3   predicate_section: "Predicates:" predicate_definition+
4   premise_section: "Premises:" premise+
5   conclusion_section: "Conclusion:" conclusion+
6
7   predicate_definition: PREDICATE "(" VAR ("," VAR)* ")" COMMENT  -> define_predicate
8   premise: quantified_expr COMMENT -> define_premise
9   conclusion: quantified_expr COMMENT -> define_conclusion
10
11  quantified_expr: quantifier VAR "(" expression ")" | expression
12  quantifier: "{forall}" -> forall | "{exists}" -> exists
13
14  expression: bimplication_expr
15
16  ?bimplication_expr: implication_expr ("{iff}" bimplication_expr)?  -> iff
17  ?implication_expr: xor_expr ("{implies}" implication_expr)?  -> imply
18  ?xor_expr: or_expr ("{xor}" xor_expr)?                     -> xor
19  ?or_expr: and_expr ("{or}" or_expr)?                 -> or
20  ?and_expr: neg_expr ("{and}" and_expr)?            -> and
21  ?neg_expr: "{not}" quantified_expr               -> neg
22          | atom
23
24  ?atom: PREDICATE "(" VAR ("," VAR)* ")" -> predicate
25      | "(" quantified_expr ")"
26
27  // Variable names begin with a lowercase letter
28  VAR.-1: /[a-z][a-zA-Z0-9_]*/  | /[0-9]+/
29
30  // Predicate names begin with a capital letter
31  PREDICATE.-1: /[A-Z][a-zA-Z0-9]*/
32
33  COMMENT: /:::.*\n/
34
35  %import common.WS
36  %ignore WS
```

*Listing 8.* Prover9 Grammar

*Table 3.* Comparison of CRANE and baselines on GSM-Symbolic based on accuracy, no. of tokens, and average time.

| Model | k | Method | Acc. (%) | Parse (%) | Tokens |
|---|---|---|---|---|---|
| Qwen2.5-1.5B-Instruct | 2 | Unconstrained w/o CoT | 20 | 98 | 18.23 |
| | | Constrained | 21 | 95 | 34.28 |
| | | Unconstrained CoT | 22 | 90 | 130.74 |
| | | **CRANE** | **28** | 96 | 140.52 |
| Qwen2.5-1.5B-Instruct | 4 | Unconstrained w/o CoT | 18 | 95 | 18.23 |
| | | Constrained | 18 | 96 | 34.28 |
| | | Unconstrained CoT | 24 | 94 | 130.74 |
| | | **CRANE** | **30** | 98 | 140.52 |
| Qwen2.5-1.5B-Instruct | 8 | Unconstrained w/o CoT | 21 | 97 | 23.34 |
| | | Constrained | 22 | 97 | 25.29 |
| | | Unconstrained CoT | 26 | 90 | 128.97 |
| | | **CRANE** | **31** | 100 | 131.3 |
| Qwen2.5-Coder-7B-Instruct | 2 | Unconstrained w/o CoT | 37 | 96 | 17.22 |
| | | Constrained | 36 | 99 | 18.61 |
| | | Unconstrained CoT | 32 | 84 | 148.87 |
| | | **CRANE** | **37** | 96 | 155.65 |
| Qwen2.5-Coder-7B-Instruct | 4 | Unconstrained w/o CoT | 36 | 96 | 16.89 |
| | | Constrained | 36 | 100 | 18.81 |
| | | Unconstrained CoT | 35 | 89 | 151.29 |
| | | **CRANE** | **37** | 97 | 163.21 |
| Qwen2.5-Coder-7B-Instruct | 8 | Unconstrained w/o CoT | 36 | 94 | 17.92 |
| | | Constrained | 35 | 99 | 25.28 |
| | | Unconstrained CoT | 37 | 88 | 138.38 |
| | | **CRANE** | **39** | 94 | 155.32 |
| Qwen2.5-Math-7B-Instruct | 2 | Unconstrained w/o CoT | 20 | 66 | 115.22 |
| | | Constrained | 26 | 95 | 26.99 |
| | | Unconstrained CoT | 28 | 72 | 190.51 |
| | | **CRANE** | **32** | 89 | 195.65 |
| Qwen2.5-Math-7B-Instruct | 4 | Unconstrained w/o CoT | 22 | 83 | 47 |
| | | Constrained | 29 | 98 | 27.08 |
| | | Unconstrained CoT | 28 | 76 | 184.35 |
| | | **CRANE** | **37** | 88 | 194.77 |
| Qwen2.5-Math-7B-Instruct | 8 | Unconstrained w/o CoT | 27 | 89 | 25.7 |
| | | Constrained | 29 | 99 | 26.81 |
| | | Unconstrained CoT | 29 | 82 | 155.26 |
| | | **CRANE** | **38** | 94 | 158.86 |
| Llama-3.1-8B-Instruct | 2 | Unconstrained w/o CoT | 19 | 61 | 157.36 |
| | | Constrained | 23 | 95 | 45.58 |
| | | Unconstrained CoT | 29 | 84 | 198.64 |
| | | **CRANE** | **35** | 94 | 206.85 |
| Llama-3.1-8B-Instruct | 4 | Unconstrained w/o CoT | 18 | 68 | 131.5 |
| | | Constrained | 24 | 96 | 37.38 |
| | | Unconstrained CoT | 26 | 92 | 172.21 |
| | | **CRANE** | **30** | 97 | 179.95 |
| Llama-3.1-8B-Instruct | 8 | Unconstrained w/o CoT | 21 | 73 | 128.38 |
| | | Constrained | 26 | 98 | 35.97 |
| | | Unconstrained CoT | 30 | 95 | 163.55 |
| | | **CRANE** | **33** | 95 | 170.22 |

*Table 4.* Comparison of CRANE and greedy and sampling baselines with different models on GSM-Symbolic.

| Model | Method | pass@1/2/3 (%) | Parse (%) | Tokens |
|---|---|---|---|---|
| | Unconstrained w/o CoT (Greedy) | 21 | 97 | 23.34 |
| | Unconstrained w/o CoT (t = 0.7) | 15/19/22 | 88/96/98 | 20.19/39.76/60.57 |
| | Constrained (Greedy) | 22 | 97 | 25.29 |
| Qwen2.5-1.5B-Instruct | Unconstrained CoT (Greedy) | 26 | 90 | 128.97 |
| | Unconstrained CoT (t = 0.7) | 21/25/30 | 78/91/96 | 146.22/292.96/444.61 |
| | **CRANE** | **31** | 100 | 131.3 |
| | Unconstrained w/o CoT (Greedy) | 21 | 73 | 128.38 |
| | Unconstrained w/o CoT (t = 0.7) | 15/21/25 | 51/74/84 | 106.88/232.75/369.86 |
| | Constrained (Greedy) | 26 | 98 | 35.97 |
| Llama-3.1-8B-Instruct | Unconstrained CoT (Greedy) | 30 | 95 | 163.55 |
| | Unconstrained CoT (t = 0.7) | 24/29/**35** | 89/98/98 | 196.01/403.68/607.7 |
| | **CRANE** (Greedy) | 33 | 95 | 170.22 |

*Table 5.* Comparison of CRANE and greedy and sampling baselines with different models on FOLIO.

| Model | Method | pass@1/2/3 (%) | Compile (%) | Tokens |
|---|---|---|---|---|
| | Unconstrained CoT (Greedy) | 36.95 | 70.94 | 350.64 |
| | Unconstrained CoT (t = 0.7) | 16.75/28.57/34.98 | 35.96/55.67/68.47 | 401.5/800.19/1219.33 |
| Qwen2.5-7B-Instruct | Constrained (Greedy) | 37.44 | 87.68 | 775.62 |
| | **CRANE** (Greedy) | **42.36** | 87.68 | 726.88 |
| | Unconstrained CoT (Greedy) | 32.02 | 57.14 | 371.52 |
| | Unconstrained CoT (t = 0.7) | 14.29/22.66/29.06 | 33.99/46.8/57.64 | 435.35/877.33/1307.45 |
| Llama-3.1-8B-Instruct | Constrained (Greedy) | 39.41 | 86.21 | 549.75 |
| | **CRANE** (Greedy) | **46.31** | 85.71 | 449.77 |

