# OpenReview forum: "CRANE: Reasoning with constrained LLM generation"
_ICML.cc/2025/Conference — ICML 2025 poster_

### Official Review · Reviewer_6Prs · 2025-03-12

**Overall Recommendation:** 4

**Summary:**

The authors prove that constrained LLM generation diminishes the capabilities of
LLMs. The (not precise) essence of this result is that
1. Under constrained decoding, logspace-uniform threshold circuits (in $TC^0$)
   can obtain the same outputs (solve the same problems) as LLMs.
2. Under unconstrained decoding, LLMs can simulate Turing machines (in $NL$).
Because these two computational models belong to different complexity classes,
it follows that there are problems (like st-connectivity) which LLMs can solve
in an unconstrained setting and cannot solve in a constrained setting, (unless
$TC^0 = NL$).

The second part establishes that augmenting the grammar to allow unconstrained
decoding for reasoning (then following by the constrained decoding) allows one
to obtain constrained output and simulate the same Turing machine as an
unconstrained LLM.

To bring this result to the realm of practical applicability, the paper proposes
CRANE, a method to interleave constrained and unconstrained LLM decoding using
an augmented grammar. Parts that require constrained decoding are generated
between specific symbols (like << and >>), and there can be multiple such parts.

## Update after rebuttal

I would like to thank the authors for their insightful comments. I will maintain my score.

**Claims And Evidence:**

The theoretical results are proven. The empirical results are strong and
demonstrate the effectiveness of the proposed CRANE algorithm.

**Essential References Not Discussed:**

I think the RICHES framework (https://arxiv.org/abs/2407.00361v1) is somewhat
related in that it interleaves unconstrained decoding with exact retrieval keys,
and the present work interleaves unconstrained and constrained decoding.

**Experimental Designs Or Analyses:**

I'm not sure what the ablation study on few-shot examples ablates, I don't see anything that's ablated.

**Methods And Evaluation Criteria:**

The benchmarks are appropriate and state-of-the-art.

**Other Comments Or Suggestions:**

- this part could be made more exact, it sounds like one can choose whether to use to
  use just unconstrained or just constrained strategies: "CRANE is compatible
  with various decoding strategies, both constrained and unconstrained"
- Typo in Proposition 3.1: thershold
- This could be rephrased: "This approach allows a simple and cost-efficient approach"
- Typo on line 373: comparisong

**Other Strengths And Weaknesses:**

I really liked the discussion about the limitations.

**Questions For Authors:**

In Proposition 3.3, there is just one reasoning part before the single
constrained decoding part. In CRANE, these two are freely interleaved with
multiple constrained decoding parts. Do the results in Proposition 3.3
generalize naturally to this case? Also, is it always true that the final answer
is at the end? Why?

The paper says that CRANE works with an arbitrary LLM and constrained decoding
algorithm. How do these generate the start and end delimiters? I have seen that
the delimiters are in the prompt, but doesn't the grammar of the constrained
generation also need to be modified to include the delimiters, like in C.5.1?

**Relation To Broader Scientific Literature:**

The paper is related to the literature of constrained decoding. The theoretical
part is also related to complexity theory and the relationship between LLMs and
computational models like boolean circuits or the Turing machines.

**Theoretical Claims:**

I couldn't verify the proofs of the theoretical claims.

---

> ### Author Rebuttal · Authors · 2025-04-01
>
> Dear reviewer 6Prs,
>
> Thanks for your constructive feedback.
>
> > Q1. In CRANE, these two are freely interleaved with multiple constrained decoding parts. Do the results in Proposition 3.3 generalize naturally to this case? Is it always true that the final answer is at the end?
>
> **R1:** Proposition 3.3 shows that an ideal LLM (i.e., with a perfect set of parameters) can use constrained decoding with the augmented grammar $G_a$ (without any unconstrained steps) to simulate the steps of a given Turing machine and produce the final output only at the end. This result also naturally extends to CRANE, which interleaves unconstrained and constrained generation using $G' \to S_1GS_2$, where $G$ is the output grammar and $S_1, S_2$ are delimiters. The only modification needed in the proof is to configure the ideal LLM from Proposition 3.3 to wrap the final output in delimiters $S_1$ and $S_2$. In this case, the ideal LLM with CRANE can simulate the Turing machine’s steps and write the answer at the end. We will include a formal proof in the revised version of the paper.
>
> Practical LLMs may not have the specific set of parameters for the theoretical construction as in the ideal LLM and may not strictly follow this ideal format, where the output is generated only at the end. Instead, they may generate multiple strings that are parsable within the grammar (e.g., syntactically valid mathematical expressions). In such cases, following the common practice for CoT [1,2,3], for both CRANE and unconstrained reasoning, we greedily select the last parsable string to compute the functional accuracy. It is possible that the final answer is not generated at the end, and better selection algorithms could further improve the functional accuracy of CRANE. However, we leave this for future work.
>
> Nevertheless, our experiments demonstrate that CRANE, even with this simple selection strategy, outperforms direct final answer generation (whether in constrained or unconstrained modes) and unconstrained reasoning.
>
> [1] "Large Language Models are Zero-Shot Reasoner",  Advances in Neural Information Processing Systems 35 (NeurIPS 2022)\
> [2] "GSM-Symbolic: Understanding the Limitations of Mathematical Reasoning in Large Language Models", The Thirteenth International Conference on Learning Representations (ICLR 2025)\
> [3] https://huggingface.co/datasets/apple/GSM-Symbolic#answer-extraction-heuristic
>
>
> > Q2. Doesn't the grammar of the constrained generation also need to be modified to include the delimiters, like in C.5.1?
>
> **R2:** The constrained decoding algorithm must be initialized with the augmented grammar that includes the delimiter symbols. We mentioned this in the pseudocode (Algorithm 1, lines 3 and 4). The grammars in C.5.1 also include the delimiter symbols. For example, line 916 of the GSM-Symbolic grammar includes `<<` and `>>`. We will update the paper to clarify this detail.
>
> > Q3. Typos and writing suggestions.
>
> **R3:** Thanks for pointing this out. We will fix them in revised version of the paper.

---

> > ### Comment · Reviewer_6Prs · 2025-04-03
> >
> > Thank you for your insightful answers! I will maintain my score.

---

### Official Review · Reviewer_ssfH · 2025-03-14

**Overall Recommendation:** 3

**Summary:**

The paper introduces CRANE (Constrained Reasoning Augmented Generation), a decoding algorithm for grammar-constrained large language model (LLM) generation that aims to balance syntactic correctness with reasoning capabilities. The work first provide a theoretical explaination for why constrained generation diminish the expressive power of LLMs. It then proposes an augmented grammar soltuion that allow intermediate reasoning steps to occur unrestricted first before restricted generation. The approach is demonstrated to improve both syntactic correctness and functional correctness.

## update after rebuttal
The rebuttal addresses my questions. I remain positive on the paper.

**Claims And Evidence:**

+ Claims are generally well-supported by theoretical propositions and empirical evaluation. Specifically,
the work shows improvements on two benchmark datasets compared to baseline approaches.

**Essential References Not Discussed:**

+ the work comprehensively cites relevant prior works.
- Many reasoning approaches are proposed to allow LLMs tackle more complex tasks. The work could discuss how these reasoning-related work affect the proposed approach.

**Experimental Designs Or Analyses:**

The experimental designs seem sound. The proposed approach is compared to relevant approaches on two benchmarks using multiple LLM models.

**Methods And Evaluation Criteria:**

+ The evaluation methods are appropriate. The work evaluate its approach on two symbolic reasoning datasets (GSM-symbolic, and FOLIO) and uses evaluation tools such as Z3 solver and Prover9 for validation the correctness.

**Other Comments Or Suggestions:**

Minor typo:

line 373: comparisong against --> comparison against
line 498: Opneai tools --> OpenAI tools

**Other Strengths And Weaknesses:**

+ clear theoretical contributions
+ simple yet effective approaches to improve grammar-constrained LLM generation
- focusing on grammars with finite languages seems to significantly limit the applicable scope of the work as the heavyweight usage of LLM is on code generation for common (e.g., rust) and domain-specific languages, which are likely all infinite. This could be the reason why the work focuses on the selected two benchmarks.

**Questions For Authors:**

1. Does the choice of the delimiter symbols (S1 and S2) affect the performance of CRANE? Currently, it is << and >> for the delimeters. Have the authors try other symbol choices?

2. The models evaluated are all relatively small. Have the authors try larger models which could have better reasoning capability to begin with? Does the proposed approach see diminishing gain?

3. Can the authors explain how would advances in reasoning models affect the grammar-constrained generation? In particular, the effectiveness of CRANE?

**Relation To Broader Scientific Literature:**

The work clearly relates to prior works on constrained decoding.
It provides two main key contributions based on its position:
1. The theoretical analysis explaining the limited reasoning capability from constrained generation
2. A simple yet effective approach to tradeoff between constrained generation and reasoning.

**Theoretical Claims:**

+ The author presents clearly the limitation of prior work on providing only empirical studies. This work instead provide theoretical studies on why constrained generation limits the reasoning capability of the LLM. The theoretical analysis seems sound.

---

> ### Author Rebuttal · Authors · 2025-04-01
>
> Dear reviewer ssfH,
>
> Thanks for the constructive feedback. We have included additional experiments with different delimiters and larger and newer reasoning models. All experiments below use the same setup as Section 5. In all cases, CRANE consistently outperformed the baselines. We will add these results in the revised version.
>
> > Q1. Experiments with different delimiters.
>
> **R1:** We use `<<` and `>>` as the delimeter for GSM-Symbolic following the original paper (see Figure 1) [1]. We will make this detail more clear in the revised version of the paper.
>
> Furthermore, to highlight CRANE's effectiveness across different delimeters, we have performed an ablation study on the GSM-symbolic dataset. We use ``## ##`` and ``!! !!`` in place of `<< >>` and report comparison against unconstrained CoT, which always performed the best among all three baselines.
>
> | Delimeter | Model | Method | Acc. (%) | Parse (%) | Tokens |
> | -------- | -------- | -------- |-------- |-------- | -------- |
> | || Unconstrained CoT | 20 | 61 | 156.13 |
> | |Qwen2.5-1.5B-Instruct | **CRANE** | **23** | 96 | 165.24 |
> | || Unconstrained CoT | 27 | 70 | 164.29 |
> | ``## ##``| Qwen2.5-Math-7B-Instruct | **CRANE** | **35** | 93 | 162.73 |
> | || Unconstrained CoT | 23 | 80 | 150.34 |
> | |Qwen2.5-1.5B-Instruct | **CRANE** | **26** | 92 | 162.21 |
> | || Unconstrained CoT | 24 | 81 | 198.53 |
> | ``!! !!``| Qwen2.5-Math-7B-Instruct | **CRANE** | **32** | 89 | 255.79 |
> | | | | | | |
>
> > Q2. Experiments with larger and advanced reasoning models.
>
> **R2:** We add experiments for newly released reasoning and larger models on GSM-symbolic. CRANE consistently outperforms the baselines on all models.
>
>
> | Model | Method | Acc. (%) | Parse (%) | Tokens |
> | -------- | -------- | -------- |-------- |-------- |
> | | Unconstrained w/o CoT | 18 | 89 | 21.64 |
> | | Constrained | 20 | 99 | 17.21 |
> | | Unconstrained CoT | 24 | 89 | 212.24 |
> | DeepSeek-R1-Distill-Qwen-7B | **CRANE** | **29** | 92 | 235.78 |
> | | Unconstrained w/o CoT | 12 | 77 | 29.2 |
> | | Constrained | 13 | 96 | 16.89 |
> | | Unconstrained CoT | 21 | 87 | 250.83 |
> | DeepSeek-R1-Distill-LLaMA-8B | **CRANE** | **31** | 92 | 268.82 |
> | | Unconstrained w/o CoT | 42 | 93 | 18.52 |
> | | Constrained | 42 | 96 | 25.62 |
> | | Unconstrained CoT | 42 | 95 | 157.71 |
> | Qwen2.5-Coder-14B-Instruct | **CRANE** | **45** | 95 | 158.54 |
> | | Unconstrained w/o CoT | 29 | 82 | 20.9 |
> | | Constrained | 30 | 91 | 30.48 |
> | | Unconstrained CoT | 32 | 87 | 233.42 |
> | DeepSeek-R1-Distill-Qwen-14B | **CRANE** | **38** | 93 | 244.98 |
> | | Unconstrained w/o CoT | 37 | 80 | 54.93 |
> | | Constrained | 38 | 91 | 34.2 |
> | | Unconstrained CoT | 43 | 87 | 222.62 |
> | QwQ-32B | **CRANE** | **46** | 88 | 237.98 |
> | | | | | |
>
> > Q3. Advances in Reasoning Models and Grammar-Constrained Generation
>
> **R3:** CRANE's effectiveness is maintained with the latest reasoning models. Larger, more advanced reasoning models are still susceptible to syntactic and semantic errors. Additionally, when recent reasoning models generate lengthy reasoning traces, enforcing output constraints from the start—as done in baseline constrained decoding algorithms—significantly degrades LLM performance.
> CRANE addresses both of these challenges by dynamically enforcing constraints. Notably, with DeepSeek-R1-Distill-LLaMA-8B, CRANE achieves the highest performance gains, improving by 10%  points.
> Therefore, for future reasoning models that generate reasoning traces and require final results to provably satisfy structural constraints, CRANE remains a valuable tool.
>
>
> > Q4. Finite Grammars.
>
> **R4:** Both the gsm-symbolic and FOLIO grammars permit infinite strings. The experiments demonstrate that CRANE significantly enhances the performance of practical LLMs, even when the output grammar allows infinite strings. Finite output grammars $G$ are just a special case used to theoretically establish the negative result, highlighting the limitations of constrained decoding when using highly restrictive grammars for **any** constant-layer LLM. Overall, CRANE remains a valuable framework for improving the performance of practical LLMs, even with infinite output grammars.
>
> On the other hand, Proposition 3.3 already provides a positive result for the ideal setting. It shows that an **ideal LLM** (with perfect parameter values) can always simulate any Turing machine using constrained decoding alone. This is possible when the output grammar $G$ is augmented with additional rules, as done in $G_a$. However, the currently available LLMs may not possess the perfect set of parameters.
>
> > Q5. Typos.
>
> **R5:** Thanks for pointing this out. We will fix them in the revised version.
>
> [1] "GSM-Symbolic: Understanding the Limitations of Mathematical Reasoning in Large Language Models", ICLR, 2025.

---

### Official Review · Reviewer_jTKi · 2025-03-14

**Overall Recommendation:** 3

**Summary:**

The paper examines the expressivity of large language models (LLMs) under grammar constraints. It first demonstrates that there are problems LLMs can solve without constraints but fail to solve when under grammar constraints. The paper introduces the CRANE algorithm, which first generates an unconstrained reasoning chain and then imposes constraint grammar within specific generated symbols.

**Claims And Evidence:**

The claims are sound and well-supported.
- The paper claims that an LLM with a constant number of layers cannot solve all problems. The authors first give an example of a st-connectivity problem, which is NL-complete but has constraints of single output digit 0 or 1. Then, they prove that the LLM can not solve it with the constraint unless the logspace-uniform class TC0=NL.
- The experiments can validate the effectiveness of the proposed algorithm. The results show the proposed method significantly outperforms all baselines with different LLMs while achieving a comparable grammar pass rate with constrained grammar.

**Essential References Not Discussed:**

The paper cites sufficient related works.

**Experimental Designs Or Analyses:**

The experimental designs are sound. The paper evaluates the method on Math and Logic datasets and compares it with baseline with constraints and COT.

**Methods And Evaluation Criteria:**

The paper uses accuracy and the grammatically correct rate to evaluate the methods. It is sound to inspect whether the methods follow the required grammar.

**Other Comments Or Suggestions:**

n/a

**Other Strengths And Weaknesses:**

Weakness:
- The paper provides a theoretical explanation of why COT is essential for LLMs across various tasks. However, COT has recently become a widely adopted technique. The proposed method integrates COT with output constraints and thus has limited novelty.

**Questions For Authors:**

n/a

**Relation To Broader Scientific Literature:**

The theoretical claims may inspire further works on LLM expressivity under constraints and design inference methods for certain grammars.

**Theoretical Claims:**

The proofs of Proposition 3.1 and 3.3 are correct.

---

> ### Author Rebuttal · Authors · 2025-04-01
>
> Dear reviewer jTKi,
>
> Thanks for your constructive feedback.
>
> > Q1. The paper provides a theoretical explanation of why COT is essential for LLMs across various tasks.  However, COT has recently become a widely adopted technique. The proposed method integrates CoT with output constraints and thus has limited novelty.
>
> **R1:** Although Chain-of-Thought (CoT) is a popular technique, to the best of our knowledge, we are the first to formally show that constrained decoding with restrictive grammars can limit the ability of large language models (LLMs) to perform CoT. This, for the first time, provides a theoretical justification for earlier empirical evidence [1] showing that the functional accuracy of LLMs for certain tasks decreases under constrained decoding. More importantly, we show that the lack of expressive power arises only when the output grammar is overly restrictive, and this limitation can be easily mitigated by augmenting the output grammar with additional production rules for any given task. This insight leads to our proposed adaptive decoding algorithm, CRANE, which improves performance over both baseline constrained and unconstrained decoding methods.
>
> Our primary contribution is not just the combination of CoT with constrained decoding but:
> 1. Identifying and formally establishing the potential limitations of constrained decoding with restrictive grammars for any LLM, including more advanced models developed in the future.
> 2. Providing both -- a theoretical augmented grammar construction for any task (Turing Machine) and a simple, practical algorithm—CRANE—that simultaneously preserves reasoning capabilities and reduces structural errors (syntax and semantic errors) in generated outputs, thereby enhancing overall LLM performance.
>
> [1] "Let Me Speak Freely? A Study on the Impact of Format Restrictions on Performance of Large Language Models", Proceedings of the 2024 Conference on Empirical Methods in Natural Language Processing: Industry Track.

---

### Decision · Program_Chairs · 2025-05-01

**Decision:**

Accept (poster)

**Comment:**

This paper first posts a theoretical analysis on the observation that grammar-guided constraint-decoding for LLMs would reduce the reasoning capabilities, by proving that the computational model under constraint and unconstraint decoding belong to different complexity class, thus there must be problems that can be solved with unconstraint decoding that inherently can not be solved with constraint decoding. To mitigate this issue, it proposes CRANE, a decoding algorithm that interleave constrained and unconstrainted LLM decoding with the augmented grammar. Experiments are conducted on GSM-Symbolic (math) and FOLIO (logical reasoning domains), and show that the proposed method is effective in improving the reasoning capabilities of LLMs. The theoretical analysis on constraint decoding is valuable and the experiments are well-designed to showcase the effectiveness of the proposed CRANE method. Concerns on generalizability to larger models are adequately addressed during the rebuttal phase by experimenting on large models. The remaining concern of this work is relatively limited scope for the experiments. As the main goal is to improve LLM reasoning in general and the theoretical analysis is also towards general LLM reasoning, the experiments are only conducted on two domain (math and logical reasoning), and only one dataset for each domain. I would suggest the authors to augment the experiments with more domains, or at least more datasets for each domain (e.g., there are plenty other math reasoning datasets out there) to provide better empirical support.